# Influence of Iron Addition (Alone or with Calcium) to Elements Biofortification and Antioxidants in *Pholiota nameko*

**DOI:** 10.3390/plants10112275

**Published:** 2021-10-24

**Authors:** Sylwia Budzyńska, Marek Siwulski, Zuzanna Magdziak, Anna Budka, Monika Gąsecka, Pavel Kalač, Piotr Rzymski, Przemysław Niedzielski, Mirosław Mleczek

**Affiliations:** 1Department of Chemistry, Faculty of Forestry and Wood Technology, Poznań University of Life Sciences, Wojska Polskiego 75, 60-625 Poznań, Poland; zuzanna.magdziak@up.poznan.pl (Z.M.); monika.gasecka@up.poznan.pl (M.G.); miroslaw.mleczek@up.poznan.pl (M.M.); 2Department of Vegetable Crops, Faculty of Agriculture, Horticulture and Bioengineering, Poznań University of Life Sciences, Dąbrowskiego 159, 60-594 Poznań, Poland; marek.siwulski@up.poznan.pl; 3Department of Mathematical and Statistical Methods, Poznań University of Life Sciences, Wojska Polskiego 28, 60-637 Poznań, Poland; anna.budka@up.poznan.pl; 4Department of Applied Chemistry, Faculty of Agriculture, University of South Bohemia, 370-04 České Budějovice, Czech Republic; kalac@zf.jcu.cz; 5Department of Environmental Medicine, Poznań University of Medical Sciences, Rokietnicka 8, 60-806 Poznań, Poland; rzymskipiotr@ump.edu.pl; 6Integrated Science Association (ISA), Universal Scientific Education and Research Network (USERN), Rokietnicka 8, 60-806 Poznań, Poland; 7Faculty of Chemistry, Adam Mickiewicz University in Poznań, Uniwersytetu Poznańskiego 8, 61-614 Poznań, Poland; pnied@amu.edu.pl

**Keywords:** metal interaction, functional food, iron deficiencies, mushroom supplementation

## Abstract

Mushrooms supplementation with iron (Fe) is usually limited, and therefore it would be beneficial to search for other vital elements able to improve the process. The aim of this study was to verify a possible interaction between Fe and calcium (Ca) to estimate the role of the addition of the latter metal to stimulate Fe accumulation in *Pholiota nameko*. Additionally, an analysis of phenolic compounds and low molecular weight organic acids (LMWOAs) was performed. The increase of Fe concentration in the substrate caused a significantly higher accumulation of this metal in *P. nameko*. The addition of Ca (5 or 10 mM) stimulated Fe accumulation, just as Fe concentration in the substrate stimulated Ca accumulation, which pointed to a synergism between these metals. The obtained results show that the presence of Fe in the substrate may also promote K, Mg, Mn, Na, P, and S accumulation. In contrast, the addition of Ca stimulates and/or inhibits their content in fruit bodies. The phenolic and organic acids profile was poor. Only gallic, 4-hydroxybenzoic, sinapic and syringic acids (phenolics), as well as citric and succinic acids (LMWOAs), were quantified in some combinations in *P. nameko* fruiting bodies.

## 1. Introduction

Nowadays, no one doubts that our survival and health depends largely on the quality and safety of the food we eat. An imbalance between the macro and micronutrients required for metabolism reactions leads to malnutrition. One of its forms is undernutrition resulting from nutrient deficiencies (these terms are often incorrectly used interchangeably) [1]. Unfortunately, elemental deficiencies are still a relevant health issue in developing and underdeveloped countries [2]. The causes of undernutrition or even hidden hunger include deficiency of nutrients in soils, calcareous, or alkali reactions, mono-cropping in agriculture, insufficient quantitative and qualitative supply of food, as well as allergies and intolerances [3,4].

A potentially nutrient improving intake strategy for enriching our diet with specific elements is food fortification [5]. Methods that offer meaningful solutions include classical, industrial, large-scale fortification and enhancement through genetic engineering [4,6]. Another approach is biofortification. This involves plant breeding and production to enhance the product’s nutritional value by supplementing it with bioavailable nutrients present in the natural human diet in an inadequate amount [7].

One of the chemical elements characterized by a deficit due to low bioavailability is iron (Fe). This element is a versatile transition metal found in almost all living organisms on Earth and one of the essential micronutrients required by plants and animals [8]. In humans, this fundamental trace element is used to synthesize heme, iron–sulfur containing proteins and other vital cofactors in many enzymes involved in respiration, redox reactions, catalysis, synthesis, and transcription of DNA. Unfortunately, although Fe is one of the most abundant elements, its availability is limited in soil. It exists mainly as insoluble ferric hydroxides and thus is bio-unavailable [9,10]. Hence, plants are a limited dietary Fe source, which is crucial because most people rely on plant-based foods as their primary metal source [8].

Iron deficiency (ID) is one of the leading contributors to the worldwide burden of disease. It mainly affects children, premenopausal women, and people in low- and middle-income countries [11]. It is one of the major nutritional problems worldwide as it affects approximately one-third of the world population. Fortification offers a chance to counteract this issue. Clinical studies have shown that consuming Fe-fortified foods is one of the most effective ways to prevent the element deficiency [12]. Therefore, there are many literature data on the production of food supplemented by Fe, for example, cereal [13], drinking water [14], yoghurt [15], and salt [16].

Nevertheless, the development of efficacious Fe-fortified dietary products has proven more challenging than developing nutritionally effective foods supplemented with other micronutrients. Unlike most other elements, Fe can cause unacceptable changes to foods. For example, the more bioavailable soluble compounds cause inappropriate colour and flavour changes, whereas less soluble Fe compounds cause less sensory changes but are much less well absorbed [17]. Therefore, it is essential to continuously innovate the processing technologies and novel ingredients for metal fortification. Mushrooms can be an interesting object of supplementation. They are attractive, healthy food because they are poor in calories and fat but rich in proteins, minerals, and dietary fibre. Promising anticarcinogenic, antigenotoxic, and immunomodulatory activities of polysaccharides and other macromolecules derived from mushrooms are another factor in their favour [18,19,20]. Due to their high element accumulation capacity, bio-enriched mushrooms can serve as food supplements rich in antioxidants and sources of macro- and micro-elements in fortified food production [21]. Additionally, the verification of potential interactions between the supplemented elements [22] seems to be an exciting issue that gives hope for increasing the efficiency of this process.

Given the above, we decided to look at the Fe fortification of *Pholiota nameko*. The species was selected for the most promising results for Fe supplementation during the preliminary experiment. The aim of this study was to determine the role of calcium (Ca) addition to Fe-fortification and verify a possible interaction between these elements in *P. nameko* fruit bodies with an analysis of phenolic compounds and low molecular weight organic acids (LMWOAs).

## 2. Materials and Methods

### 2.1. Microorganisms and Spawn

The PN04 strain of *P. nameko* was used in the experiment. The strain was obtained from the Mushroom Collection of Poznań University of Life Science (Department of Vegetable Crops). The spawn for inoculation of substrates was prepared by the method described by [23].

### 2.2. Substrate Preparation

A substrate derived from a mixture of alder and beech sawdust (1:1 vol.) for cultivation was used in the experiment. It was additionally supplemented with 20% wheat bran and 5% cornmeal. The substrate was mixed with LPM 20 stirrer (Glass, Germany) and moistened with Fe (iron (III) chloride hexahydrate) and Ca (calcium nitrate hydrate) salts dissolved in distilled water. The salts of both metals were used in this way to obtain final concentrations in the substrate, which were: 5, 10, and 15 mM of Fe and 5 and 10 mM of Ca in the following systems: Fe5, Fe10, Fe15, Fe5+Ca5, Fe10+Ca5, Fe15+Ca5, Fe5+Ca10, Fe10+Ca10, and Fe15+Ca10. As a control system, substrate without the addition of Ca and Fe salts was used.

After mixing with the solutions, the substrates had a humidity of 60%. They were then placed in polypropylene bottles of 0.85 dm^3^ volume. Each bottle was filled with 450 g of the substrate and closed with PP filter (class F-9, Filtropol, Poland). The substrates were sterilized at 121 °C for one hour, and next were cooled down to 25 °C, after which they were inoculated with 10 g of spawn (on wheat grain) for each bottle.

Incubation was conducted at a temperature of 25 °C and 80–85% air relative humidity for 30 days. The bottles with removed covers and bags were then placed in the cultivation room. For fructification, air relative humidity was maintained at 95–100% and temperature at 15 ± 1 °C. The cultivation was additionally lighted with fluorescent light of 500 lx intensity for 12 h a day. The growth room was aerated in such a way as to maintain CO_2_ concentration below 1000 ppm. Yield included whole fruit bodies. Depending on the combination, the pins appeared after 7–10 days, and the fruiting bodies grew to harvest within the next 5–6 days. The fruiting bodies were harvested by twisting them out of the substrate (whole clump), and then the remnants of the substrate were removed. Whole fruiting bodies constituted the yield. Harvesting was performed when most of the fruiting bodies in a clump had a broken veil covering the gills, but the cap was not yet fully open. One flush of fruiting bodies was collected. The yield was calculated as the mean of five replicates. It was the mass per one bottle (i.e., 450 g of the substrate).

The growth of the mycelium was also studied using the same media. The substrates were filled in glass tubes (18 × 2 cm) to a height of 15 cm, sealed with cellulose plugs and then sterilized in the same conditions as those in bottles. The substrates were inoculated by placing a 1 cm layer of mycelium on wheat grains on the upper surface of the substrates. Incubation in tubes was carried out at 25 °C. The measure of mycelium growth was the layer of the substrate overgrown with hyphae after 28 days of incubation (see Figure 1).

### 2.3. Studies on Total Concentration of Elements

#### 2.3.1. Procedure

The dry samples (dried at 45 ± 5 °C in an electric oven Thermocenter, Salvislab, Switzerland) were accurately weighed 0.200–0.500 g (±0.001 g) and then digested by nitric acid (65%; Sigma-Aldrich, St. Louis, MO, USA) in closed Teflon containers at 180 °C (20 min ramp time, 20 min hold time, 20 min cooling downtime) in the microwave digestion system Mars 6 Xpress (CEM, Matthews, NC, USA). After digestion, samples were diluted with water (purified in MilliQ water purification system Millipore, Darmstadt, Germany) to a total volume of 15.0 mL.

#### 2.3.2. Instrumentation

The inductively coupled plasma high-resolution optical emission spectrometer ICP-hrOES PlasmaQuant 9100 Elite (Analytik Jena, Jena, Germany) was used for calcium (Ca), iron (Fe), magnesium (Mg), sodium (Na), potassium (K), manganese (Mn), phosphorus (P), and sulphur (S) determination. Different wavelengths and plasma observations were used for low and high element concentrations (except Mn-Appendix A). The common conditions were applied: axial plasma view for low and radial plasma view for high concentration, Radio Frequency (RF) power 1.20 kW, plasma gas flow 12.0 L min^−1^, nebulizer gas flow 0.50 L min^−1^, auxiliary gas flow 0.5 L min^−1^, and sample flow rate 0.65 mL min^−1^; the signal was measured in five replicates during 1 s each. The calibration curves were constructed based on four points separately for the low and high concentrations of elements using different wavelengths (Appendix A). Argon (Ar) was used as an internal standard (emission at wavelength 420.068 nm was controlled).

#### 2.3.3. Analytical Method Validation

The detection limit was determined at the level of 0.01 mg kg^−1^ dry weight (DW) based on 3-sigma criteria (Appendix A). Accuracy was checked by analysis of the reference materials NCSDC (73349)—bush branches and leaves; CRM S-1—loess soil; CRM 667—estuarine sediments; CRM 405—estuarine sediments; CRM 2709—soil; and the recovery (80–120%) was acceptable. The uncertainty for the total analytical procedure (including sample preparation) was at the level of 20%.

### 2.4. Determination of Phenolic Compounds and Organic Acids

Phenolic compounds and organic acids were extracted from homogenized fruit bodies of *P. nameko* (5 g) using 80% ethanol (25 g). The samples were sonicated at 40 °C for 20 min (Bandelin Sonorex RK 100, Berlin, Germany), shaken for 12 h at room temperature and then centrifuged at 2700 g for 15 min at 25 °C (Universal 320R Hettich Zentrifugen, Tuttlingen, Germany) evaporated to dryness and stored at −20 °C before analyses.

The determination was carried out using ultra-performance liquid chromatography, ACQUITY UPLC H-Class System (Waters Corp., Milford, MA, USA), with separation on an Acquity UPLC BEH C18 column (2.1 mm × 150 mm, 1.7 μm, Waters, Milford, MA, USA) thermostated at 35 °C. The mixture of water and acetonitrile (both containing 0.1% formic acid, pH = 2) at the flow rate of 0.4 mL min^−1^ with the gradient elution was used (flow 0.4 mL/min—5% B (2 min), 5–16% B (5 min), 16% B (3 min), 16–20% B (7 min), 20–28% B (11 min) flow 0.45 mL/min—28% (1 min), 28–60% B (3 min) flow 5.0 mL/min—60–95% B (1 min), 65% B (1 min), and 95–5% B (0.1 min) flow 0.4 mL/min—5% B (1.9 min) and the injection volume was 5 μL. The identification of peaks was based on a comparison with the retention times of chemical standards. The Waters Photodiode Array Detector (Waters Corporation, Milford, MA, USA) was applied for the detection of individual components at λ = 280 nm and λ = 320 nm [24].

### 2.5. Determination of Total Phenolic (TP) Content

A mixture of 200 µL of the extracts and 1 mL of Folin-Ciocalteu reagent (diluted with deionized water v:v (1:1) was incubated for 3 min, then 1 mL of 20% Na_2_CO_3_ was added. The samples were kept in darkness for 30 min at room temperature. The absorbance of the samples was measured at 765 nm using a UV-spectrophotometer. Gallic acid was used as the standard for TP content quantification. The concentration of total phenolic content (TP) was expressed as milligram gallic acid equivalents per dried weight (mg GAE · g^−1^ DW). Gallic acid was used as the standard for TP quantification [25].

### 2.6. Statistical Analysis

Statistical analyses were performed using the agricole package (R) (Bell Laboratories). To compare mean content of elements, phenols, and LMWOAs in *P. nameko* growing in particular experimental systems, one-dimensional analysis of variance (ANOVA), and finally, the multiple comparisons Tukey’s HSD test were performed, which allowed the existence of the uniform groups of objects to be shown (α = 0.05). For a graphical presentation of the relationships between particular objects (*P. nameko* fruit bodies in experimental systems) due to the content of elements, phenols, and LMWOAs, Principal Component Analysis (PCA) was performed [26]. Heatmaps were prepared to show similarities/differences between mushrooms in particular experimental systems regarding all elements, antioxidants, and studied parameters jointly [27]. All data on chemical elements’ content in the text and tables are given in mg kg^−1^ DW. For each condition and analysis, five replicates were determined.

## 3. Results

### 3.1. Mycelium Growth and Mushroom Yield

The mycelium in the control grew the fastest. In each case, when the addition of Fe and Ca was applied, the mycelium grew more slowly. However, it was found that inhibition of mycelium growth depended on the dose of Fe and Ca. The mycelium grew the most slowly with the addition of Fe and Ca in the highest concentration. In this case, the synergistic effect of these metals was noticeable. It was found that the higher the Fe dose, the slower the mycelial growth was (Figure 1).

The addition of Fee alone or in combinations with Ca led to the negative effect on the weight (amount) of the fruiting bodies (on yield). Fe (5 and 10 mM) did not cause a significant decrease of yield (76.0 and 69.9 g, respectively) concerning the control (78.7 g) (Figure 2). However, the addition of 15 mM of Fe led to a significant decrease in *P. nameko* yield (47.9 g). A similar tendency was observed for the rest of the experimental systems, where under 5 or 10 mM of Ca, the addition of 5 (64.4 and 56.7 g, respectively) or 10 (54.0 and 44.5 g, respectively) mM of Fe a similar yield was observed, becoming significantly higher than after 15 mM of Fe and 5 or 10 mM of Ca addition (44.5 and 17.6 g, respectively). The general decrease in the fruit body yield was not accompanied by any changes in their colour and only slight modifications in cap shape (Figure 3).

### 3.2. Mineral Composition of Mushroom Bodies

#### 3.2.1. Content of Iron and Calcium

The average content of Fe in the control *P. nameko* was significantly lower (8.61 mg kg^−1^) than after the addition of this metal to the substrate (5, 10 and 15 mM). It was: 15.9, 19.8 and 25.1 mg kg^−1^, respectively (Figure 4a). The addition of 5 mM of Ca to the substrate caused a significantly higher accumulation of Fe (23.1, 24.7, and 31.6 mg kg^−1^, respectively for Fe_5_Ca_5_, Fe_10_Ca_5_, and Fe_15_Ca_5_) (Appendix A). It is worth underlining that the addition of 10 mM of Ca to 15 mM of Fe caused the highest accumulation of this metal (38.3 mg kg^−1^), which was clearly observed in PCA analysis, where 61.99% (36.47 + 25.52) of total variability was explained.

The content of Ca in the control variant of *P. nameko* and the Fe_5_, Fe_10_ and Fe_15_ experimental systems was almost the same (233, 241, 219, and 225 mg kg^−1^, respectively) (Figure 4b). The addition of 5 mM of Ca and an increase of Fe content in the substrate (Fe_5_Ca_5_, Fe_10_Ca_5_, and Fe_15_Ca_5_) was related to significantly higher Ca content than in the control (458, 698, and 931 mg kg^−1^, respectively). Moreover, the addition of 10 mM of Ca caused an even higher accumulation of Ca (825, 1430, and 1630 mg kg^−1^, respectively), also confirmed by PCA analysis, where Ca was situated between Fe_10_Ca_10_ and Fe_15_Ca_10_. The obtained results indicate that the mutual interaction between these metals and their concentration in the substrate affects their accumulation in *P. nameko* fruit bodies.

#### 3.2.2. Content of Other Elements

The addition of 15 mM of Fe only significantly influenced higher K accumulation by *P. nameko* fruit bodies than the control (19,100, and 11,300 mg kg^−1^, respectively) (Figure 5a). The addition of Ca (5 or 10 mM) did not lead to a significantly higher/lower content of K in mushrooms compared to the control. Increasing the Fe concentration in substrate stimulated Mg accumulation by *P. nameko* bodies (651, 852, and 928 mg kg^−1^, respectively, for the Fe_5_, Fe_10_, and Fe_15_ systems), similarly to the presence of Ca (5 mM) (614, 870, and 937 mg kg^−1^, respectively, for the Fe_5_Ca_5_, Fe_10_Ca_5_, and Fe_15_Ca_5_ systems) (Appendix A). Content of Mg in mushrooms growing in substrate enriched with 10 mM of Ca was significantly higher than for control and almost the same independently of Fe addition (724, 709, and 728 mg kg^−1^, respectively for the Fe_5_Ca_10_, Fe_10_Ca_10_, and Fe_15_Ca_10_ systems) (Figure 5b). The content of Mn in control mushrooms was 6.71 mg kg^−1^, while the addition of Fe or Fe + Ca significantly stimulated the accumulation of this metal (Figure 5c). The highest content of Mn in mushrooms growing under the Fe_10_Ca_10_ and Fe_15_Ca_10_ systems was determined (29.5 and 28.2 mg kg^−1^, respectively) and clearly visible in the PCA analysis (Figure 6).

Interesting results were observed for Na content in mushrooms (Figure 5d). Control *P. nameko* contained 104 mg kg^−1^, similar to Fe_5_Ca_5_ and Fe_5_Ca_10_ (114 and 123 mg kg^−1^, respectively). The addition of 5, 10, and 15 mM of Fe caused a significantly higher content of Na in mushrooms than for the control (130, 173, and 183 mg kg^−1^, respectively). The presence of Ca (5 or 10 mM) did not influence Na accumulation except for Fe_10_Ca_5_ and Fe_10_Ca_10_, where a significantly lower content of Na was determined (153 mg kg^−1^ for both systems) than in mushrooms exposed to the Fe_10_ system (173 mg kg^−1^) (Appendix A).

*Pholiota nameko* growing in substrate enriched with 10 or 15 mM of Fe contained a significantly higher amount of P (7230 and 7430 mg kg^−1^, respectively) than in the control (6640 mg kg^−1^) (Figure 5e). The addition of 5 mM of Ca to substrate did not influence P content in fruit bodies, while the addition of 10 mM of Ca inhibited P accumulation in two experimental systems (Fe_5_Ca_10_ and Fe_15_Ca_10_) (Appendix A). The growth of mushrooms in substrate enriched with Fe or Fe + Ca caused a significantly higher content of S than in the control (3600 mg kg^−1^). The highest content of S was recorded in *P. nameko* growing under Fe_15_Ca_5_, Fe_10_Ca_5_, Fe_15_Ca_10_, and Fe_15_ (4330, 4270, 4110, and 4190 mg kg^−1^, respectively) (Figure 5f).

A heatmap was prepared to show the similarity between *P. nameko* fruit bodies growing in a particular experimental system (Figure 7). Generally, three groups of objects were identified: 1st) Control, Fe_5_, Fe_5_Ca_5_, and Fe_5_Ca_10_; 2nd) Fe_10_Ca_10_ and Fe_15_Ca_10_; and 3rd) Fe_10_, Fe_15_, Fe_10_Ca_5_, and Fe_15_Ca_5_ with a similarity between the last two groups. The obtained results pointed to the significant role of Fe and Ca addition concerning the content of all determined elements jointly. Moreover, a lower addition of Fe and Ca in the substrate influenced the generally lower content of other elements. In contrast, mushroom exposure to higher contents of Fe and Ca tended to have the opposite effect.

### 3.3. Profile of Phenolic Compounds

The profile of phenolic compounds varied in all combinations (Table 1). The lowest and highest content of phenolic compounds was confirmed for syringic acid in the control and Fe_15_Ca_10_, respectively (9.18 and 50.8 μg g^−1^ DW).

In the control, only gallic and syringic acids were identified. The most distinct phenolic acid profile was observed for mushrooms cultivated on the substrate with Fe and addition Fe_5_Ca_5_. In comparison to the control, 4-hydroxybenzoic acid (4-HBA) was detected in mushrooms cultivated in Fe and Fe_5_Ca_5_-supplemented substrates. Gallic acid was quantified in the control and mushrooms cultivated in Fe and Fe_5_Ca_5_-enriched substrates. Sinapic acid was detected only for Fe_5_Ca_10,_ while syringic was found in almost all combinations. The simultaneous supplementation of the substrate with Fe and Ca in most combinations did not result in the detection of 4-HBA and gallic acid with the significant increase of syringic acid content.

TP content was significantly affected by the supplementation of substrates. The lowest TP content, 0.81 mg g^−1^ GAE, was found in the control. The addition of Fe to substrate resulted in a significant increase of TP up to 1.44 mg g^−1^ GAE as the higher content for Fe_15_. The supplementation of the substrate by Fe and Ca also increased TP content in comparison to the control. However, no significance in TP content was confirmed in any samples supplemented with Fe and Ca.

### 3.4. Profile of Low Molecular Weight Organic Acids

The profile and content of low molecular weight organic acids in the present experiment was very poor (Table 2). In general, only citric and succinic acids were identified in *P. nameko* fruiting bodies. The addition of Fe to the substrate (Fe_5_, Fe_10_, and Fe_15_ systems) only caused the creation of succinic acid in *P. nameko* bodies, and at a lower content compared to the control. Additionally, compared to the control, the fruiting bodies did not contain citric acid. For the Fe_5_Ca_10_, Fe_10_Ca_10_, and Fe_15_Ca_5_, systems, where Ca and Fe were added simultaneously, the amount of succinic acids observed significantly increased (8.83, 28.6, and 36.4 μg g^−1^ DW ^1^, respectively) and the creation of citric acid was stimulated in *P. nameco* fruit bodies (1.30, 1.47, and 1.63 μg g^−1^ DW ^1^, respectively). In addition, acetic acid was determined in the Fe_10_Ca_10_ system, while it was below the detection level for the other systems.

## 4. Discussion

Although the problem has been well known for over 50 years, ID remains the most common cause of anaemia worldwide and affects approximately 1–2 billion people of all age groups [28,29]. The major factors are reduction or complete absence of metal intake; an agricultural revolution that resulted in animal foods rich in bioavailable Fe being displaced by cereals, legumes, and plant-based diets; as well as the poor condition of the soils in which the consumed plants grow [30]. The critical issue is the bioavailability of Fe, and it has been estimated to be in the range of 14–18% for mixed diets and 5–12% for vegetarian diets. Dietary factors that influence Fe absorption, such as ascorbic acid, Ca intake, phytate, polyphenols, and muscle tissue, have been identified repeatedly [31]. Hence, there is need to search for new sources of the element, including fortification products.

It has been confirmed that mushrooms can bioaccumulate several metals of nutritional and pharmacological importance [32,33,34]. More and more reports of attempts to supplement mushrooms with Fe can also be found in the literature. Almeida et al. [35] showed cultivation variables to increase Fe bioaccumulation in *Pleurotus ostreatus* mycelium. The effect of culture medium modifications: Fe (50 or 150 mg L^−1^), carbon (glucose or carboxymethyl cellulose), nitrogen (hydrolyzed casein or yeast extract), and pH (4.5 or 6.5) have been evaluated. Yokota et al. [36] examined *P. ostreatus* cultivated on substrate supplemented with various Fe concentrations (from 50 to 5000 mg kg^−15^ DW, which correspond to from 25 to 2500 mg L^−1^). Biological efficiency, metal content, bioavailability, the content of ashes, proteins and antioxidant activity have been determined. Umeo et al. [37] considered the bioaccumulation capacity of *Agaricus subrufescens* grown on Fe enriched substrate (50 mg L^−1^). The authors described mycelial biomass and the concentration of bioaccumulated Fe. The research team of Meniqueti et al. [38] also investigated Fe fortification in the mushroom species *A. subrufescens*, *Ganoderma lucidum*, *P. eryngi*, *P. ostreatus*, and *Schizophyllum commune* under different metal doses (10–100 mg L^−1^). Our experiment described the effect of 5, 10, and 15 mM Fe (corresponding to 280, 560, and 840 mg L^−1^) added to the mushroom medium. Compared to the additives used by other authors, these are relatively high concentrations.

A common observation for mentioned in the previous paragraph studies was the reduction of the biomass obtained in the presence of Fe addition. In our study, biomass decreased to 97, 89, and 61% (for 5, 10, and 15 mM Fe) compared to the control. Almeida et al. [35] also reported a strong growth decrease in *P. ostreatus* for Fe doses above 150 mg L^−1^ and no growth at 300 mg L^−1^. Significant growth reduction with Fe addition above 300 mg L^−1^ and total inhibition above 750 mg L^−1^ was described for *P. ostreatus* by Yokota et al. [36]. What is more, from the 14 *A. subrufescens* strains tested by Umeo et al. [37], 12 were characterized by lower biomass production when 50 mg L^−1^ Fe was added (ranging from 19 to 85% compared to the control). A similar observation was made by Meniqueti et al. [38], where 100 mg L^−1^ Fe doses caused a reduction in the mycelial biomass of *A. subrufescens*, *P. eryngii*, *P. ostreatus*, and *S. commune* (43, 6, 37 and 52% of the control). However, it should be mentioned that in the same experiment, *G. lucidum* and *P. eryngii* exposed to 50 and 30 mg L^−1^ Fe, respectively, showed 1.3-fold more biomass than the respective controls.

It is not surprising that the increased concentration of elements in the medium is usually associated with a more significant accumulation in mushrooms. In our study, mean Fe content in supplemented *P. nameko* was from almost 185 to 292% of the control, for Fe from 5 to 15 mM added to the substrate, respectively. The crucial fact is that all the research cited above [35,36,37,38] mentioned similar results.

It is essential to look at different approaches to improve Fe contents and bioavailability from fortified foods. It seems promising to study the interactions between individual nutrients added to the medium in search of favourable intake and synergistic effects [39]. In our experiment, we added 5 and 10 mM of Ca (which correspond to 200 and 400 mg L^−1^ Ca) as a modulating factor for Fe accumulation. The addition of the second element contributed to a further reduction in biomass production compared to systems supplemented only with Fe. For a dose of 5 mM of Fe, the addition of 5 and 10 mM Ca resulted in 85 and 75%; for a dose of 10 mM Fe, 77 and 64% and for a dose of 15 mM Fe, 93 and 37% biomass compared to the only Fe-fortified ones, respectively.

Most importantly, the supplementation of Ca to the medium increased the accumulation of Fe by the tested mushrooms. The addition of 5 mM Ca resulted in 145, 125, and 126% of Fe accumulation concerning mushrooms grown on the medium with 5, 10, and 15 mM Fe addition only. Also, adding 10 and 15 mM Ca to Fe-fortified substrate resulted in a higher accumulation of Fe than for systems with only Fe addition. The highest increase, 153% of Fe content in mushrooms growing on 15 mM Fe, was observed when 10 mM Ca was simultaneously added to the substrate. Although we could not find any literature data on similar mushroom supplementation of Fe with the addition of Ca, we can try to find common conclusions from the coincident supplementation of other elements. Scheid et al. [39] explored Fe biofortification and availability of fortification *Lentinus crinitus*, *G. lucidum*, *S. commune*, *P. ostreatus*, *P. eryngii* and *Lentinula edodes*. Mushrooms were grown on malt-extract or sugarcane molasses with different concentrations of Fe (0.116 and 91.23 mg L^−1^, respectively) as well as Mg (260.0 and 903.1 µg L^−1^, respectively). The obtained observations are a component of all the factors that differed in both systems. Nevertheless, it can be concluded that for five from seven species, the medium with higher Fe and Mg concentrations showed higher mycelial biomass growth, and for all tested species, mushrooms grown on this medium accumulated significantly higher contents of Fe. The differences in Mg concentration probably influenced these observations. Interestingly, studies of the effect of the addition of Mn to Fe-fortified substrate on the growth and accumulation of *L. crinitus* have been published by Meniqueti et al. [40]. The authors tested the effect of Mn (0.9 mg L^−1^) on mushroom growth, bioaccumulation, and transfer of Fe at different cultivation times (7, 14, and 21 days) in a culture medium. Iron and Mn added to the substrate did not biomass affect growth. Manganese increased Fe bioaccumulation by up to two-fold when compared to Fe-fortification only.

Phenolic compounds are responsible for many functions which are connected with the ability to scavenge free radicals with a chelating ability to ions. In food, phenolic compounds are advisable because they increase the consumption value of the product. The phenolic composition of *P. nameko* has not been widely studied. The few studies that are available indicate that the composition is very poor [41,42], however TP content was similar to other species. The obtained results also show that the profile of phenolic compounds in *P. nameko* is rather poor; only four phenolic acids were quantified. Supplementation with Fe has a very favorable effect on the synthesis of 4-HBA and causes a significant increase in the acid. Some mixtures of Fe and Ca in different concentrations affects the elevation of synthesis of phenolic acids. Supplementation modified the profile and initiated the synthesis of some new components in comparison to the control, and significantly increased TP content. In the earlier studies on the supplementation of *P. nameko* substrates with Se salts, a significant increase in individual phenolic compounds and TP was achieved [41]. The increase in TP content in mushrooms was confirmed for Se-enrichment [33,43,44]. Mineral enrichment is recognized as a method of promoting the increase of health benefits of food. However, in *P. nameko*, enriched by Fe and Ca, no spectacular changes in phenolic composition were recorded. Similar results were obtained by Fontes Vieira et al. [45] on *P. ostreatus* enriched with Fe which showed no differences in TP between enriched and non-enriched mushrooms. The effect of enrichment of substrates by different elements on biosynthesis is still not recognized. Changes in the phenolic content of enriched mushrooms are probably the result of activation or deactivation of the biochemical pathway at various stages, which may have been influenced by the type of salt in enriched substrate.

Organic acids are important constituents of mushroom taste components, which have a close relation to synthesis of phenolic compounds, amino acids, esters, and metabolic process of aroma components [44,45]. However, their most important function, besides phenolic compound synthesis, is that organic acids are known to possess antioxidative properties [46,47] having specific health effects, and even though they are non-nutritive compounds, some of them have been detected in different fungal species [48]. Moreover, the currently published literature data shows that the existence of a different profile and content of the determined acids in similar fruit bodies, of both wild and cultivated species, results from the occurrence of a diverse growth environment (specific climate combined with microhabitual characteristic) [49]. In *P. nameko*, the content of organic acids is limited, and only two dominant acids (succinic and citric) were identified, which from the entire spectrum of these compounds constitute a small percentage. Supplementation with Fe led to the lower synthesis of succinic acid in comparison to the control, while the mixture of Fe and Ca in different concentrations resulted in its elevation, and the simultaneous creation of citric acid. Supplementation modified the profile and initiated the synthesis of citric acids and significantly increased the content in comparison to the control. Succinic acid has been found in fruit bodies of various species of fungi such as *L. fumosum*, *L. gilva*, and *L. scabrum* [48], as well as in *L. edodes*, for which a significant increase was found when using a substrate with a high content of high C/N values [44].

Furthermore, research results indicate that regular consumption of edible mushrooms in the human diet or their nutritional supplements can provide health benefits by improving antioxidant defense mechanisms and reducing the risk of chronic diseases [49,50]. Possible mechanisms of the antioxidant activity of mushroom species can be attributed to their strong hydrogen donation capacity or their effectiveness as good free radical scavengers, including the organic acids analyzed in our study [24,51]. Nevertheless, the presented study showed that *P. nameko* cannot be used as a readily available source of strong natural antioxidants and antimicrobials or as a viable dietary supplement, or in the food and pharmaceutical industries due to the fact that the amount of organic acids in the fruiting bodies was very limited.

The results showed that supplementation of *P. nameko* by Fe and Ca does not improve the phenolic composition of fruiting bodies. The modification of phenolic composition, the content of each compound, TP and organic acids has little impact. However, increasing the Fe content in fruit bodies may be a cost-effective method of improving the quality of the fruiting bodies for the consumer. Recommended dietary Fe intake for adult females and males has been estimated as 9–15 and 6–11 mg day^−1^, respectively [52,53]. Assuming the standard dose of mushroom consumption as 25 g DW per day, it can be calculated that a product from the Fe_15_ system will provide less than 5% and 7%, while one from the Fe_15_Ca_10_ system will provide almost 8% and 11% of the daily requirement for Fe, for females and males, respectively. At the same time, an increased Ca content can be an added value. Recommended dietary Ca intake for an adult male has been estimated as 900–1200 mg day^−1^ [54,55]. Mushrooms from the Fe_15_ system will provide less than 1%, while those from the Fe_15_Ca_10_ system will provide 4% of the daily requirement of Ca.

## 5. Conclusions

Despite the enormous advancement in knowledge about food and nutrition, ID is one of the leading contributors to the worldwide burden of disease. Due to the fact that ID is common, additional sources of this element should be sought. Mushrooms can be a valuable food for Fe supplementation, but some modifications are necessary. Utilizing the effect of the interaction of other elements with Fe during supplementation on increasing the amount of metal intake is highly promising. The supplementation with Fe and simultaneous addition of Ca described in our study could be a starting point for planning the practical application of such cultivation. Of course, further research requires a reliable assessment of the bioavailability of Fe in such mushrooms in the gastrointestinal tract in order to assess to what extent the fruiting bodies will satisfy the demand for Fe and also Ca due to its accumulation. There is also no doubt that above conclusions will only be relevant if, from the point of view of the economics of mushroom growing, the decisive parameter will not be the amount of harvest, but the quality of product.

## Figures and Tables

**Figure 1 plants-10-02275-f001:**
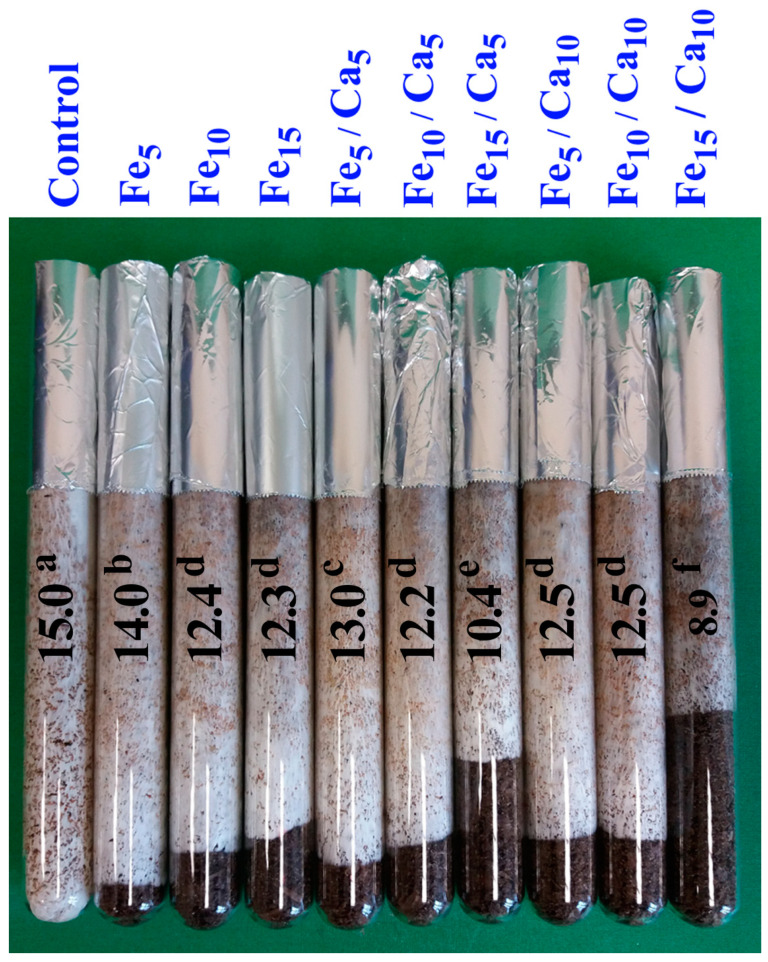
Characteristics of mycelium growth (cm) for particular experimental systems. *n* = 5, identical lower cases (a, b…) denote non-significant differences between mean *Pholiota nameko* growth in particular experimental systems according to post-hoc Tukey’s HSD test.

**Figure 2 plants-10-02275-f002:**
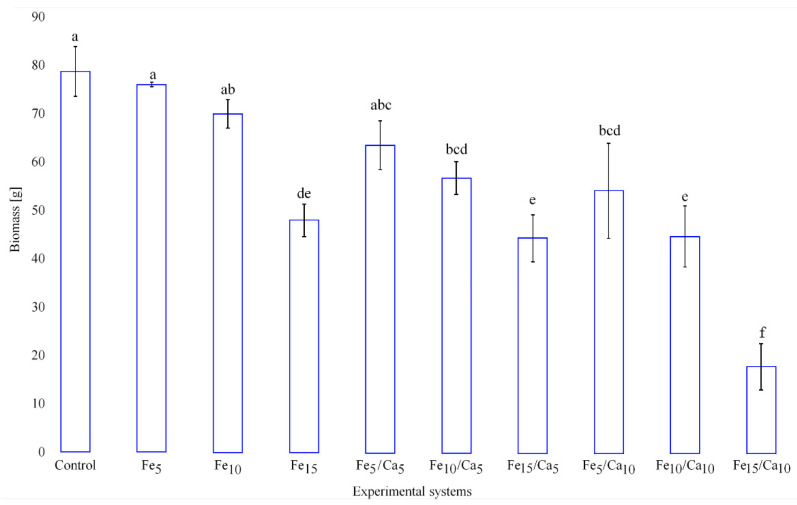
Yield (g) of *Pholiota nameko* exposed to particular experimental systems. *n* = 5, identical lower cases (a, b…) denote non-significant differences between mean *Pholiota nameko* yield growing in particular experimental systems according to post-hoc Tukey’s HSD test.

**Figure 3 plants-10-02275-f003:**
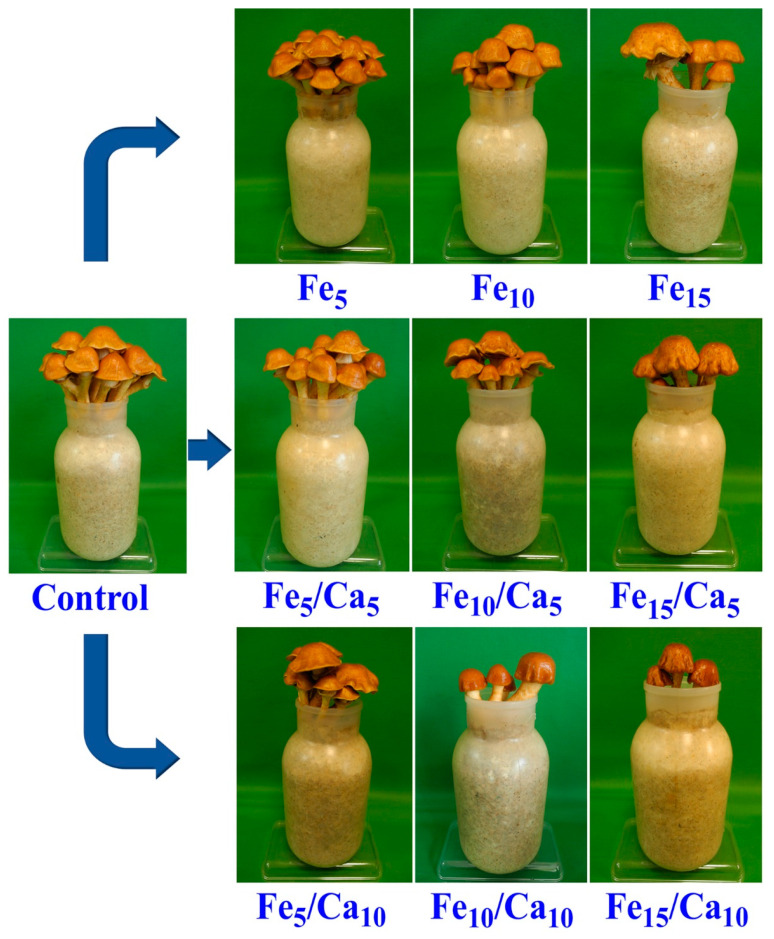
Macroscopic characteristics of *Pholiota nameko* exposed to particular experimental systems.

**Figure 4 plants-10-02275-f004:**
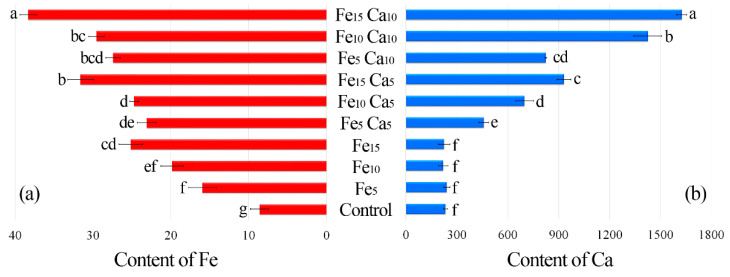
Content of Fe (**a**) and Ca (**b**) (mg kg^−1^ DW) in mushrooms exposed to particular experimental systems. *n* = 6, identical lower cases (a, b…) denote non-significant differences between mean content of Ca or Fe in *Pholiota nameko* fruit bodies growing in particular experimental systems according to post-hoc Tukey’s HSD test.

**Figure 5 plants-10-02275-f005:**
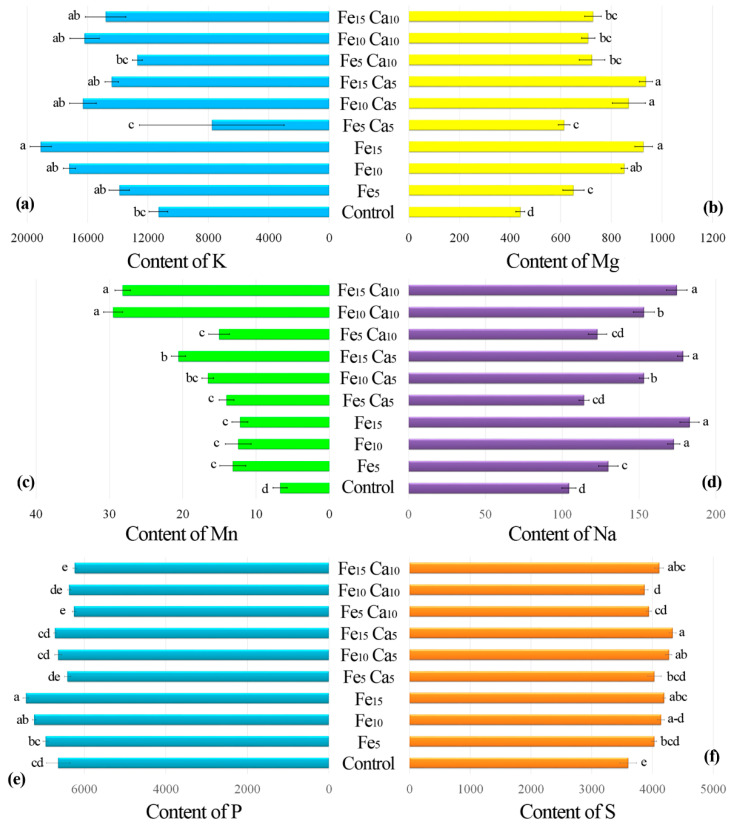
Content of K (**a**), Mg (**b**), Mn (**c**), Na (**d**), P (**e**), and S (**f**) (mg kg^−1^ DW) in mushrooms exposed to particular experimental systems. *n* = 5, identical lower cases (a, b…) denote non-significant differences between mean content of particular metals in *Pholiota nameko* fruit bodies growing in particular experimental systems according to post-hoc Tukey’s HSD test.

**Figure 6 plants-10-02275-f006:**
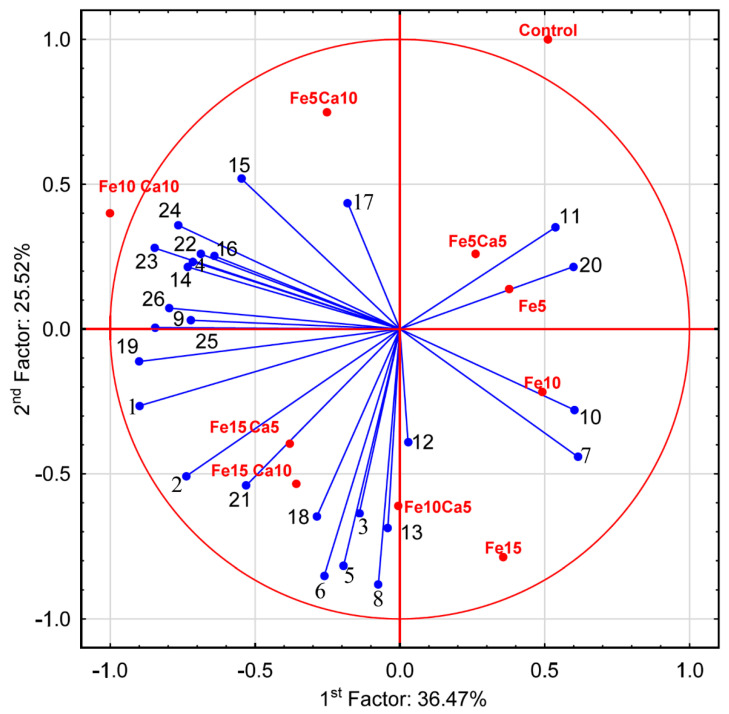
Principal Component Analysis for parameters characterizing mushrooms exposed to particular experimental systems. 1−Ca; 2−Fe; 3−K; 4−Mg; 5−Mn; 6−Na; 7−P; 8−S; 9−protocatechuic; 10−4—hydroxybenzoic acid; 11−catechin; 12−vanillic acid; 13−syringic acid; 14−2,5 dihydroxybenzoic acid; 15−caffeic acid; 16−chlorogenic acid; 17−sinapic acid; 18−TP; 19−quinic acid; 20−malic acid; 21−lactic acid; 22−citric acid; 23−acetic acid; 24−fumaric acid; 25−succinic acid; and 26−sum of LMWOAs.

**Figure 7 plants-10-02275-f007:**
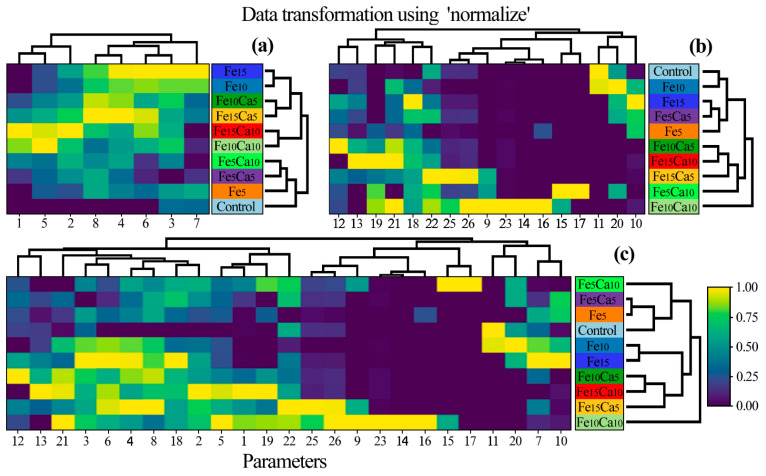
Correlation between particular *Pholiota nameko* of all experimental systems concerning the content of (**a**) studied elements, (**b**) antioxidants, and (**c**) all studied parameters (Heatmap) in mean values with presentation of a hierarchical tree plot. 1−Ca; 2−Fe; 3−K; 4−Mg; 5−Mn; 6−Na; 7−P; 8−S; 9−protocatechuic; 10−4—hydroxybenzoic acid; 11−catechin; 12−vanillic acid; 13−syringic acid; 14−2,5 dihydroxybenzoic acid; 15−caffeic acid; 16−chlorogenic acid; 17−sinapic acid; 18−TP; 19−quinic acid; 20−malic acid; 21−lactic acid; 22−citric acid; 23−acetic acid; 24−fumaric acid; 25−succinic acid; and 26−sum of LMWOAs.

**Table 1 plants-10-02275-t001:** Content (μg g^−1^ DW) of phenolic compounds.

System	4-HBA	Gallic	Sinapic	Syringic	TP [mg g^−1^ GAE *]
Control	bDL	14.8 ^ab^	bDL	9.18 ^e^	0.810 ^e^
Fe_5_	27.6 ^d^	11.1 ^c^	bDL	bDL	1.02 ^de^
Fe_10_	24.9 ^c^	13.0 ^bc^	bDL	11.7 ^de^	1.03 ^cde^
Fe_15_	37.5 ^b^	13.4 ^bc^	bDL	21.8 ^c^	1.44 ^a^
Fe_5_Ca_5_	43.2 ^a^	17.9 ^a^	bDL	11.1 ^e^	1.26 ^ab^
Fe_10_Ca_5_	bDL	bDL	bDL	31.7 ^b^	1.22 ^a–d^
Fe_15_Ca_5_	bDL	bDL	bDL	15.5 ^d^	1.16 ^bcd^
Fe_5_Ca_10_	bDL	bDL	1.78	bDL	1.19 ^bcd^
Fe_10_Ca_10_	bDL	bDL	bDL	bDL	1.17 ^bcd^
Fe_15_Ca_10_	bDL	bDL	bDL	50.8 ^a^	1.25 ^abc^

* GAE−gallic acid equivalent; *n* = 5; identical superscripts in columns denote no significant differences between means according to a post-hoc Tukey’s HSD test at α = 95% following one-way ANOVA, bDL−below detection limit, 4-HBA−4-hydroxybenzoic acid, and TP−total phenolic.

**Table 2 plants-10-02275-t002:** Content (μg g^−1^ DW) of low molecular weight organic acids.

System	Acetic	Citric	Succinic	Sum
Control	bDL	1.05 ^e^	6.89 ^cd^	7.93 ^bc^
Fe_5_	bDL	bDL	3.60 ^de^	3.60 ^cd^
Fe_10_	bDL	bDL	1.98 ^e^	1.98 ^d^
Fe_15_	bDL	bDL	4.89 ^cde^	4.90 ^bcd^
Fe_5_Ca_5_	bDL	1.19 ^d^	6.98 ^cd^	8.16 ^bc^
Fe_10_Ca_5_	bDL	bDL	6.12 ^cd^	6.12 ^bc^
Fe_15_Ca_5_	bDL	1.63 ^a^	36.4 ^a^	38.0 ^a^
Fe_5_Ca_10_	bDL	1.30 ^c^	8.83 ^c^	10.1 ^b^
Fe_10_Ca_10_	8.32	1.47 ^b^	28.6 ^b^	38.3 ^a^
Fe_15_Ca_10_	bDL	1.10 ^e^	5.39 ^cde^	6.46 ^bc^

*n* = 5; identical superscripts in columns denote no significant differences between means according to a post-hoc Tukey’s HSD test at α = 95% following one-way ANOVA, bDL−below detection limit.

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
