# Peer review of "Influence of Iron Addition (Alone or with Calcium) to Elements Biofortification and Antioxidants in Pholiota nameko"

_plants, 2021, doi:10.3390/plants10112275_

Round 1
Reviewer 1 Report
Comments to paper Plants
Influence of calcium addition to iron biofortification and anti-oxidants in Pholiota nameko
Interesting subject, food fortification, the results show a good promise to launch a new product in the future market. Several topics need to be explained as well as figures and tables must be made clearer.
Line 110, 112, 115, 127: Please relace ºC by oC
Item 2.4 and 2.5: font type changed
Line 153: instead or rpm for spin condition please use «g»
Lone 158: the gradient conditions are missing
Line 166: should be here in first place the TPC meaning, not in the next line.
Line 168: FW?
Line 171: LMWOAs: the meaning should be indicated
Statistical Analysis should indicate how many replicates were done for each condition.
How was micellium growth measured? Lines 119-124 do not explain. By weight? Line 183 talks about «thickness» how is it measured.
Figure 1 legend: the letters resulting from the statistical comparison are not indicated in the figure
Line 195-196, refers weight, while in the beginning of the item it was talked about cm. How was the growth measured?
Line 201-203: talks about «body yield», color and cap shape and in Figure 3 was is seen is indicated as «biomass (g)». Was color and shape «transformed» into body yield? Maybe changing the sentence, a little bit would be more understandable.
Figure 4 (a) and (B) are interchanged
The yy axis of figure 4a is not seen
The xx axis does not show the units
Line 214-218. The values described cannot be followed by the reader from Figure 4.
Line 217: Talks about figure 5, Ca, Fe and PCA but figure 5 is not PCA and several metals are shown. Please check all this. And where is the PCA?
Line 269: italic is missing
Lin 269: says the Figure 6 show a heat map, but what is referred in the figure legend is PCA
Figure 7 was not referred in the text.
Table 1 and 2 are not referred in the text.
Is there an explanation for the diminishing in the fungi growth in the presence of Fe?
How can the presence of one metal influence the absorption of the other. Which is the mechanism of absorption of these metals by the mushroom? Is it known?
How dos the total phenolic content of the control compares with other mushrooms species?
Line 461: «Recommended dietary Fe intake for adult males has been estimated as 6-11 mg day-1» and for females?
Line 466: why only male again for the DDR?
General comment for the discussion: the bioavailability of iron and calcium should be analyzed. The food may have a high content in these metals and the bioavailability may be low.
Author Response
Reviewer: 1
Interesting subject, food fortification, the results show a good promise to launch a new product in the future market. Several topics need to be explained as well as figures and tables must be made clearer.
Line 110, 112, 115, 127: Please relace ºC by oC
Author response: Thank you very much for your minuteness. The units have been corrected.
Item 2.4 and 2.5: font type changed
Author response: Thank you very much for your suggestion. It has been corrected.
Line 153: instead or rpm for spin condition please use «g»
Author response: It has been corrected.
Line 158: the gradient conditions are missing
Author response: It has been added.
Line 166: should be here in first place the TPC meaning, not in the next line.
Author response: It has been added.
Line 168: FW?
Author response: It has been corrected. It should be dried weight.
Line 171: LMWOAs: the meaning should be indicated
Author response: The abbreviation was explained in the Abstract and in the Introduction sections.
Statistical Analysis should indicate how many replicates were done for each condition.
Author response: Thank you very much for your vigilance. All analyses were performed with the same number of repetitions (5). It is our lack of attention that we have entered different values in the manuscript. We may only explain a broad spectrum of research in a multidisciplinary research team, which sometimes causes minor oversight. Information has been added to the section.
How was micellium growth measured? Lines 119-124 do not explain. By weight? Line 183 talks about «thickness» how is it measured. Line 195-196, refers weight, while in the beginning of the item it was talked about cm. How was the growth measured?
Author response: Thank you for your comment. The sentence about measure has been added to the section.
Figure 1 legend: the letters resulting from the statistical comparison are not indicated in the figure
Author response: Thank you for paying attention to this cardinal mistake. The descriptions of the individual figures were incorrectly matched, and it has been rewritten.
Line 201-203: talks about «body yield», color and cap shape and in Figure 3 was is seen is indicated as «biomass (g)». Was color and shape «transformed» into body yield? Maybe changing the sentence, a little bit would be more understandable.
Author response: Thank you very much for your valuable suggestions. The descriptions of the individual figures were incorrectly matched, and it has been rewritten, and the numbers of figures have been changed. Now everything should be legible.
Figure 4 (a) and (B) are interchanged
Author response: Thank you very much for your valuable observation. Figures have been changed.
The yy axis of figure 4a is not seen
Author response: Thank you very much. The y axis of both Figure 4 and 5 is one for both graphs. The way of presentation is not accidental, but it results from the fact that there is a large amount of data there. The limitation to one common description made it possible to prepare a figure with a resolution of 500 (Fig 4ab) or 400 (Fig. 5a-f) rather than 300 dpi, keeping the same size.
The xx axis does not show the units
Author response: Thank you very much. The information about the units [mg kg-1 DW] has been added to the figure descriptions. Entering the units (the same for each plot) would provide even more information, and the figure would be less understandable. It was easier to enter the unit (mg kg-1) directly into the description.
Line 214-218. The values described cannot be followed by the reader from Figure 4.
Author response: Thank you very much for your point of view. We believe that everything will depend on the final form of the figure in the publication. However, we agree that if its size is limited, it will be significantly difficult to read specific values. For this reason, we have prepared an S2 table that we have incorporated into the Supplementary material so that every reader can access and use the mean values.
Line 217: Talks about figure 5, Ca, Fe and PCA but figure 5 is not PCA and several metals are shown. Please check all this. And where is the PCA?
Author response: Thank you for paying attention to this cardinal mistake. The numbers of figures have been corrected.
Line 269: italic is missing
Author response: It has been corrected.
Line 269: says the Figure 6 show a heat map, but what is referred in the figure legend is PCA
Author response: The numbers of figures have been corrected.
Figure 7 was not referred in the text.
Author response: The numbers of figures have been corrected.
Table 1 and 2 are not referred in the text.
Author response: Thank you. It has been completed.
Is there an explanation for the diminishing in the fungi growth in the presence of Fe?
Author response: Commonly, the growth of fruiting bodies of many mushroom species under the influence of high Fe concentrations is limited. An example of such observations for Pleurotus species is the study by Ogidi et al. (2016). The authors used very high concentrations of Fe (500 and 1000 mg kg-1 of substrate), while 5, 10 and 15 mM in our work. The reason for the decrease in yield at lower concentrations may be different chemical characteristics of the substrate and interactions between the elements (deficiency or excess compared to the values considered by the fungi to be optimal). Evident changes in the content of major elements in P. nameko fruiting bodies were due to changes in the mineral profile of the substrates used in the experiment. This may also suggest interactions between Ca and N. We know that Ca addition exert a profound effect on the protein and content of soluble nitrogen, which may also cause mushroom growth limitation.
How can the presence of one metal influence the absorption of the other. Which is the mechanism of absorption of these metals by the mushroom? Is it known?
Author response: Thank you for drawing attention to this critical point. Unfortunately, the mechanism of the uptake of elements by fungi is unknown. So is choosing which elements are taken in larger amounts and others in smaller ones. It is also unclear where exactly the elements are accumulated. Nevertheless, our team have been trying to find answers to these questions for 5 years.
How dos the total phenolic content of the control compares with other mushrooms species?
Author response: Thank you very much for your suggestions. The range of TP content for many species are wild. The TP content was similar to some species.
Line 461: «Recommended dietary Fe intake for adult males has been estimated as 6-11 mg day-1» and for females? Line 466: why only male again for the DDR?
Author response: Thank you for this remark. Additional information has been added.
General comment for the discussion: the bioavailability of iron and calcium should be analysed. The food may have a high content in these metals and the bioavailability may be low.
Author response: Thank you. Further research on this issue seems to be very promising. We will try to develop these issues in the future part of our scientific work.

Reviewer 2 Report
Respect the suggestions and add answers to questions. Recommendations, suggestions and questions are also given at the edge (side) of the article.

Author Response
Respect the suggestions and add answers to questions. Recommendations, suggestions and questions are also given at the edge (side) of the article.
Author response: Thank you for your suggestions. Corrections following the comments in the pdf file have been made in the manuscript.

Reviewer 3 Report
The manuscript entitled "Influence of calcium addition to iron biofortification and anti-oxidants in Pholiota nameko" is interesting and important for the fortification of minerals in vegetables including mushrooms. There are some issues, which required to improve prior to its publication. Some them are given below:
Line 137: “S determination”….S stands for?.
Line 155: write complete address (model, city and country) of equipment “ultra-performance liquid chromatography, ACQUITY UPLC”
Line 162: Keep citation “2.5. Determination of total phenolic content”
Fig 4 and 5: Provide us more clear figures. Font sizes are small.
Line 241, 244, 269: Write the scientific name if italic form always: “P. nameko fruit bodies?
Line 304: which tree species” “Pholiota nameko of both tree species with..”
Line 313: write total phenolic content “TP-total phenolic”
Table 1: Error in statistical work.
4-HBA content in Fe5, Fe10, Fe15, Fe5Ca5 has no significantly differences?
Line 331, 470, 472: ID remains ?
Line 331: Although the problem……..” which and what problem?
340: “Hence the need to search” replace with “Hence, there is need to search..”
Line 344: repeat “Almeida et al. Al- 344 meida et al., 2015) showed cultivation variables to increase..”
Line 360: “--- these studies”…………………….which studies ? give citations.
Line 408: replace “did not biomass affect growth….. “ by “did not showed biomass affect growth…”
Line 418: “The mixture of Fe and Ca in different 418 concentrations only affects the elevation of synthesis of syringic acid.”
What kind of effect Fe and Ca has in phenolic compound biosynthesis pathway? Discuss mechanism. Use some recent literature.
Author Response
The manuscript entitled “Influence of calcium addition to iron biofortification and anti-oxidants in Pholiota nameko” is interesting and important for the fortification of minerals in vegetables including mushrooms. There are some issues, which required to improve prior to its publication. Some them are given below:
Line 137: “S determination”….S stands for?.
Author response: Thank you. The full name of S has been added.
Line 155: write complete address (model, city and country) of equipment “ultra-performance liquid chromatography, ACQUITY UPLC”
Author response: It has been completed.
Line 162: Keep citation “2.5. Determination of total phenolic content”
Author response: The citation has been added.
Fig 4 and 5: Provide us more clear figures. Font sizes are small.
Author response: The font size has been increased, especially for the description of the y axis (individual experimental setups). At the same time, the graphs were rotated in such a way that the y axis common for the 2 graphs was unambiguous. This way of presentation also made it easier to compare elements with each other.
Line 241, 244, 269: Write the scientific name if italic form always: “P. nameko fruit bodies?
Author response: Thank you. It has been corrected.
Line 304: which tree species” “Pholiota nameko of both tree species with..”
Author response: Thank you for drawing your attention to this. It was a mistake, and the description has been corrected.
Line 313: write total phenolic content “TP-total phenolic”
Author response: In all manuscript total phenolic content was codified as TP content.
Table 1: Error in statistical work.
Author response: The error has been corrected.
4-HBA content in Fe5, Fe10, Fe15, Fe5Ca5 has no significantly differences?
Author response: The mistakes have been corrected.
Line 331, 470, 472: ID remains ?
Author response: In the Introduction section full name of ID – iron deficiency was introduced. After that only ID in the whole manuscript was used.
Line 331: Although the problem……..” which and what problem?
Author response: Iron deficiency problem. We introduced the full name of ID in the Introduction section. Instead of writing every single time “iron deficiency problem”, we decide to use “ID” abbreviation and the article “the” (since this is the main problem discussed here) to avoid repetitions and unnecessarily lengthening in sentences.
340: “Hence the need to search” replace with “Hence, there is need to search..”
Author response: Thank you for your suggestion. It has been corrected.
Line 344: repeat “Almeida et al. Al- 344 meida et al., 2015) showed cultivation variables to increase..”
Author response: Thank you. It has been corrected.
Line 360: “--- these studies”…………………….which studies ? give citations.
Author response: Thank you. It has been corrected.
Line 408: replace “did not biomass affect growth….. “by “did not showed biomass affect growth…”
Author response: Thank you very much. It has also been corrected.
Line 418: “The mixture of Fe and Ca in different concentrations only affects the elevation of synthesis of syringic acid.”
Author response: The sentence was modified.
What kind of effect Fe and Ca has in phenolic compound biosynthesis pathway? Discuss mechanism. Use some recent literature.
Author response: The effects of mineral supplementation of substrates on bioactive compounds in mushrooms are weakly recognised. Thus it is very difficult to discussed this mechanism.

Reviewer 4 Report
The authors did a good work from an experimental point of view and I recommend the article for publication after some major revisions.
More specific:
L9: Use the same format for different addresses.
L21: The Abstract in one paragraph.
L126: The double parenthesis in this paragraph creates difficulties for readers.
L141: How much was the flow rate pump?
L142: …Argon (Ar) was used as an internal standard… Argon is a gas used for plasma! How did you use it as an internal standard?
L142: How many points did you use for the calibration curve from each element?
L152: How much was the solid to liquid ratio?
L152: Give more details about ultrasound. Conditions, system, etc.
L152: Was the sample stabilized at 40°C during ultrasound?
L156: There are some errors in the column details.
L163: …A mixture of the extracts and Folin-Ciocalteu reagent (diluted with deionised water v:v (1:1)… How much volume from each?
L220: In Figure 4 and 5 add the error bars.
L482: Adding information for Supplementary data.
Author Response
The authors did a good work from an experimental point of view and I recommend the article for publication after some major revisions.
More specific:
L9: Use the same format for different addresses.
Author response: Thank you very much for your suggestion. It has been corrected.
L21: The Abstract in one paragraph.
Author response: Thank you. It has also been corrected.
L126: The double parenthesis in this paragraph creates difficulties for readers.
Author response: Thank you very much for your suggestion. It has been removed.
L141: How much was the flow rate pump?
Author response: 0.65 mL min-1. The information has been added.
L142: …Argon (Ar) was used as an internal standard… Argon is a gas used for plasma! How did you use it as an internal standard?
Author response: For sequential instruments, this option is frequently used. The measurement of argon emission using a selected line (in our studies 420.068 nm) allows controlling the stability of the plasma. The information has been added to the manuscript.
L142: How many points did you use for the calibration curve from each element?
Author response: 4 points. The information has been added to the section.
L152: How much was the solid to liquid ratio?
Author response: It has been completed.
L152: Give more details about ultrasound. Conditions, system, etc.
Author response: It has been completed.
L152: Was the sample stabilised at 40°C during ultrasound?
Author response: All samples have the same weight and volume. They were placed directly in the ultrasound bath at 40ºC. Each sample was ultrasound at the same time and under the same conditions. The temperature of the samples was identical for each, and it is also increased in the same way. The temperature was also suitable for the extraction of the tested compounds, which results from numerous studies.
L156: There are some errors in the column details.
Author response: It has been corrected.
L163: …A mixture of the extracts and Folin-Ciocalteu reagent (diluted with deionised water v:v (1:1)… How much volume from each?
Author response: It has been completed.
L220: In Figure 4 and 5 add the error bars.
Author response: Thank you very much. The graphs have been slightly rebuilt and supplemented with error bars according to your suggestion. In our opinion, they are now clearer and allow easier comparison of the content of different elements in particular experimental systems. Due to the significant amount of data, we have also prepared Table S2 with average numerical values to make it easier for potential readers of this work to compare them.

Round 2
Reviewer 1 Report
The work improved a lot
Reviewer 3 Report
The authors of the manuscript revised the paper properly and responded to all the comments. Therefore, the research paper can be accepted in the present form.
Reviewer 4 Report
The paper has been revised according to the suggestions and criticisms of the reviewers. In this revised version, the paper has improved its quality and I recommend the article for publication.